# Dietary fatty acid metabolism of brown adipose tissue in cold-acclimated men

Denis P. Blondin[1], Hans C. Tingelstad[2], Christophe Noll[1], Frédérique Frisch[1], Serge Phoenix[1,3], Brigitte Guérin[3], Éric E. Turcotte[3], Denis Richard[4], François Haman[2] & André C. Carpentier[1]

In rodents, brown adipose tissue (BAT) plays an important role in producing heat to defend against the cold and can metabolize large amounts of dietary fatty acids (DFA). The role of BAT in DFA metabolism in humans is unknown. Here we show that mild cold stimulation (18 °C) results in a significantly greater fractional DFA extraction by BAT relative to skeletal muscle and white adipose tissue in non-cold-acclimated men given a standard liquid meal containing the long-chain fatty acid PET tracer, 14(R,S)-[$^{18}$F]-fluoro-6-thia-heptadecanoic acid ($^{18}$FTHA). However, the net contribution of BAT to systemic DFA clearance is comparatively small. Despite a 4-week cold acclimation increasing BAT oxidative metabolism 2.6-fold, BAT DFA uptake does not increase further. These findings show that cold-stimulated BAT can contribute to the clearance of DFA from circulation but its contribution is not as significant as the heart, liver, skeletal muscles or white adipose tissues.

[1] Department of Medicine, Centre de Recherche du Centre Hospitalier Universitaire de Sherbrooke, Université de Sherbrooke, 3001, 12e Avenue Nord, Sherbrooke, QC, Canada J1H 5N4. [2] Faculty of Health Sciences, University of Ottawa, 125 University Pvt. Ottawa, ON, Canada K1N 6N5. [3] Department of Nuclear Medicine and Radiobiology, Centre d'imagerie Moléculaire de Sherbrooke, Université de Sherbrooke, 3001, 12e Avenue Nord, Sherbrooke, QC, Canada J1H 5N4. [4] Centre de Recherche de l'Institut Universitaire de Cardiologie et de Pneumologie de Québec, Université Laval, 2725, chemin Sainte-Foy, Québec, QC, Canada G1V 4G5. Correspondence and requests for materials should be addressed to F.H. (email: fhaman@uottawa.ca) or to A.C.C. (email: andre.carpentier@usherbrooke.ca).

Disordered storage of dietary fatty acids (DFA) in white adipose tissues may lead to overexposure of lean tissues to fatty acids from triglyceride-rich lipoproteins (TRL) and nonesterified fatty acids (NEFA). This exaggerated lean tissue fatty acid exposure may ultimately result in the development of insulin resistance and impaired glucose-stimulated insulin secretion, key pathophysiological features of type 2 diabetes[1,2]. Impaired adipose tissue postprandial storage of DFA has implications for ectopic intracellular triglyceride (TG) accretion in some, but not all lean tissues[3]. Recent evidence has shown that stimulated brown adipose tissue (BAT) of mice exposed to the cold or treated with a $\beta_3$-adrenergic receptor agonist can effectively take up injected TRL-mimicking particles[4–6]. Even under thermoneutral conditions, BAT appears to mainly take up TRL-derived fatty acids[4,5], following lipoprotein lipase-mediated intravascular lipolysis, while the uptake of core remnants or whole particles also occurs but primarily with larger chylomicron-sized particles (>150 nm) and at lower environmental temperatures (7 °C versus 21 °C or 28 °C)[5]. In rodents, circulating TRL are an important source of fatty acids stored as TG in BAT[7] and may serve to the replenishment of intracellular TG stores essential for BAT thermogenesis[8,9]. The role of DFA in replenishing intracellular TG, however, remains unknown in activated BAT in humans.

The aim of the present study was to determine the role of stimulated BAT in whole-body DFA metabolism in lean healthy men before and after a 4-week cold acclimation, an intervention known to increase BAT volume and its oxidative capacity[10]. Using a novel method to noninvasively measure organ-specific DFA uptake and partitioning using oral 14(R,S)-[18F]-fluoro-6-thia-heptadecanoic acid ([18F]FTHA) and positron emission tomography (PET), we demonstrate that DFA are taken up by cold-activated BAT, but its overall contribution to systemic DFA clearance is comparatively small. Four weeks of daily cold exposure did not increase the uptake of DFA any further, despite increasing BAT oxidative capacity 2.6-fold.

## Results

**Thermal responses and metabolites unaltered by acclimation.** Eight healthy, non-cold acclimatized men aged 22 years old (95% confidence interval (CI): 20–25) with a BMI of 23.1 kg m$^{-2}$ (95% CI: 21.3–24.9) participated in the present study protocol (Table 1). During cold exposure, mean skin temperature fell by 4.6 °C (95% CI: −3.4 to −5.8) pre-acclimation and 5.6 °C (95% CI: −4.7 to −6.5) post-cold acclimation (two-way analysis of variance (ANOVA) for repeated measures with Bonferonni *post hoc* test, acute cold exposure effect: $P < 0.001$; acclimation effect: $P = 0.015$), while core temperature fell by 0.1 °C (95% CI: −0.02 to −0.22) pre-acclimation and post acclimation (Table 2). This resulted in a 1.9-fold (95% CI: 1.4–2.3) increase in energy expenditure pre-acclimation and 1.7-fold (95% CI: 1.5–1.9) post-acclimation (two-way ANOVA for repeated measures with Bonferonni *post hoc* test, acute cold exposure effect: $P < 0.001$). Both shivering activity and shivering pattern were unaltered as a result of the cold acclimation (Table 2). Plasma glucose ($Ra_{glucose}$) and plasma glycerol ($Ra_{glycerol}$) production rates increased during cold exposure/postprandial state, whereas plasma NEFA ($Ra_{NEFA}$) production rate was not significantly increased (Table 2, Supplementary Fig. 3). The postprandial area under the curve for insulin, NEFA and cortisol levels were lower post-acclimation whereas PYY and adiponectin levels increased post-acclimation (Table 2). Total plasma $^{18}F$ activity (Fig. 1a,b) as well as $^{18}F$ activity in total plasma TG (Fig. 1c,d), chylomicrons (Fig. 1e,f), very low-density lipoproteins (VLDL) (Fig. 1g,h) and NEFA (Fig. 1i,j) from the oral $^{18}FTHA$ tracer all increased over time in the postprandial phase during cold exposure (two-way ANOVA for repeated measures, $P < 0.01$), but there was no difference before compared with after the 4-week cold acclimation.

### Table 1 | Characteristics of participants.

| | Pre-acclimation | Post-acclimation | P value |
|---|---|---|---|
| Age (years) | 22 (20–25) | | |
| Weight (kg) | 73.9 (64.5–83.2) | 74.6 (65.5–83.6) | 0.11 |
| BMI (kg m$^{-2}$) | 23.1 (21.3–24.9) | 23.4 (21.7–25.0) | 0.07 |
| BSA (m$^2$) | 1.91 (1.77–1.98) | 1.92 (1.79–1.98) | 0.07 |
| Waist (cm) | 76.3 (72.1–80.5) | 76.7 (73.5–79.9) | 0.53 |
| Lean mass (kg) | 64.0 (59.5–65.8) | 63.9 (57.9–66.4) | 0.56 |
| Fasting glucose (mmol l$^{-1}$) | 4.7 (4.6–4.9) | 4.7 (4.5–4.9) | 0.73 |
| Fasting insulin (pmol l$^{-1}$) | 36.2 [32.9–40.6] | 34.6 [26.0–56.2] | 0.88 |
| Fasting C-peptide (nmol l$^{-1}$) | 0.22 [0.16–0.25] | 0.21 [0.18–0.23] | 0.83 |
| Fasting NEFA (µmol l$^{-1}$) | 402 (299–505) | 306 (190–422) | 0.17 |
| Fasting TG (mmol l$^{-1}$) | 0.6 (0.4–0.8) | 0.7 (0.6–0.8) | 0.61 |
| Fasting glucagon (ng l$^{-1}$) | 24.2 [19.4–39.8] | 28.7 [19.8–45.6] | 0.73 |
| Fasting GIP (pmol l$^{-1}$) | 8.8 (6.3–11.3) | 8.9 (6.5–11.3) | 0.93 |
| Fasting total GLP-1 (pmol ml$^{-1}$) | 9.8 (1.9–17.7) | 8.5 (1.0–16.0) | 0.79 |
| Fasting TSH (mIU l$^{-1}$) | 1.6 (1.0–2.2) | 1.6 (0.9–2.3) | 0.99 |
| Fasting free T3 (pmol l$^{-1}$) | 5.6 (4.9–6.3) | 5.5 (4.8–6.1) | 0.78 |
| Fasting free T4 (pmol l$^{-1}$) | 17.4 (16.1–18.7) | 16.7 (15.8–17.7) | 0.34 |
| Fasting ACTH (pg ml$^{-1}$) | 37.9 [28.2–100.4] | 43.8 [37.5–57.3] | 0.67 |
| Fasting cortisol (nmol l$^{-1}$) | 382 (327–438) | 328 (257–399) | 0.18 |
| Fasting leptin (ng ml$^{-1}$) | 1.1 [0.8–2.0] | 1.3 [0.7–1.9] | 0.98 |
| Fasting PYY (pmol l$^{-1}$) | 26.4 (12.8–39.9) | 30.0 (14.0–46.0) | 0.69 |
| Fasting GH (ng ml$^{-1}$) | 0.05 [0.02–0.08] | 0.03 [0.01–0.06] | 0.45 |
| Fasting adiponectin (µg ml$^{-1}$) | 3.5 (2.6–4.4) | 4.2 (2.8–5.5) | 0.33 |
| Liver volume (cm$^3$) | 1,265 (1,099–1,431) | 1,323 (1,250–1,397) | 0.39 |

NEFA, nonesterified fatty acids; TG, triglyceride.
Values are means with 95% confidence interval (CI) in parentheses for normally distributed data and median [interquartile range] for nonparametric data ($n = 8$). P values, paired-sample two-tailed *t*-test.

**Table 2 | Hormone and metabolite concentrations at room temperature and cold exposure, pre- and post-cold acclimation.**

| | Pre-acclimation | | Post-acclimation | |
|---|---|---|---|---|
| | Room temperature | Cold exposure | Room temperature | Cold exposure |
| Energy expenditure (kcal min$^{-1}$) | 1.2 [1.1–1.4] | 2.3 [1.8–2.5]**** | 1.2 [1.1–1.3] | 2.1 [1.8–2.5]**** |
| $Ts_{skin}$ (°C) | 33.3 [32.5–33.6] | 28.3 [27.9–28.7]**** | 33.2 [32.4–33.6] | 27.6 [26.8–27.9]****,# |
| $T_{core}$ (°C) | 37.1 (36.8–37.3) | 37.0 (36.7–37.2) | 37.0 (36.8–37.2) | 36.9 (36.7–37.0) |
| $\Delta T_{inlet} - T_{outlet}$ (°C) | – | 3.1 (2.5–3.8) | – | 3.0 (2.5–3.4) |
| Shivering intensity (%MVC) | – | 1.6 (0.9–2.3) | – | 2.2 (1.2–3.1) |
| Shivering burst rate (bursts min$^{-1}$) | – | 2.4 (2.0–2.8) | – | 2.6 (2.1–3.1) |
| $Ra_{glucose}$ (µmol min$^{-1}$) | 1,756 [1,705–2,056] | 2,123 [2,045–3,145]* | 1,935 [1,679–2,509] | 2,392 [2,185–2,724]* |
| $Ra_{glycerol}$ (µmol min$^{-1}$)‡ | 450 (214–686) | 904 (408–1400)** | 276 (175–377) | 963 (285–1641)** |
| $Ra_{NEFA}$ (µmol min$^{-1}$) | 597 (415–938) | 702 (576–810) | 486 [417–626] | 705 (460–762) |
| $Ox_{NEFA}$ (µmol min$^{-1}$) | 184 (159–210) | 193 (92–293) | 137 (65–208) | 221 (173–269) |
| $Ox_{DFA}$ (µmol min$^{-1}$)‡ | – | 2 (1–3) | – | 2 (1–3) |
| $AUC_{0-300}$ glucose (mmol l$^{-1}$ per 300 min) | – | 1,350 (1,224–1,476) | – | 1,267 (1,197–1,337) |
| $AUC_{0-300}$ insulin (pmol l$^{-1}$ per 300 min) | – | 40,860 (33,065–48,654) | – | 32,146 (24,396–39,896)# |
| $AUC_{0-300}$ C-peptide (nmol l$^{-1}$ per 300 min) | – | 172 [118–192] | – | 120 [91–142] |
| $AUC_{0-300}$ NEFA (µmol l$^{-1}$ per 300 min) | – | 105,431 (78,557–132,304) | – | 79,555 (63,794–95,316)# |
| $AUC_{0-300}$ TG (mmol l$^{-1}$ per 300 min) | – | 304 (222–385) | – | 319 (222–417) |
| $AUC_{120-300}$ chylomicron-TG (mmol l$^{-1}$ per 180 min) | – | 37 [16–54] | – | 31 [12–60] |
| $AUC_{120-300}$ VLDL−TG (mmol l$^{-1}$ per 180 min)‡ | – | 66 (45–86) | – | 62 (36–88) |
| $AUC_{0-300}$ glucagon (ng l$^{-1}$ per 300 min) | – | 8,724 [8,032–11,820] | – | 11,264 [9,980–13,957] |
| $AUC_{0-300}$ GIP (pmol l$^{-1}$ per 300 min) | – | 14,320 (10,446–18,194) | – | 14,762 (10,094–19,431) |
| $AUC_{0-300}$ total GLP-1 (pmol ml$^{-1}$ per 300 min) | – | 4,253 (1,095–7,410) | – | 4,533 (1,204–7,862) |
| $AUC_{0-300}$ TSH (mIU l$^{-1}$ per 300 min) | – | 372 (207–537) | – | 417 (240–595) |
| $AUC_{0-300}$ free T3 (pmol l$^{-1}$ per 300 min) | – | 1,538 (1,258–1,819) | – | 1,621 (1,444–1,797) |
| $AUC_{0-300}$ free T4 (pmol l$^{-1}$ per 300 min) | – | 4,988 (4,558–5,417) | – | 5,100 (4,797–5,402) |
| $AUC_{0-300}$ cortisol (nmol l$^{-1}$ per 300 min) | – | 96,638 (87,953–105,322) | – | 94,541 (80,806–108,277) |
| $AUC_{0-300}$ leptin (ng ml$^{-1}$ per 300 min) | – | 262 [235–442] | – | 294 [206–399] |
| $AUC_{0-300}$ PYY (pmol l$^{-1}$ per 300 min) | – | 10,290 (6,421–14,159) | – | 13,061 (7,309–18,814)# |
| $AUC_{0-240}$ GH (ng ml$^{-1}$ per 240 min) | – | 12.4 [6.0–49.0] | – | 20.5 [6.2–56.0] |
| $AUC_{0-300}$ adiponectin (µg ml$^{-1}$ per 300 min) | – | 998 (730–1,266) | – | 1,224 (858–1,590)# |

EE, energy expenditure; NEFA, nonesterified fatty acids; $Ox_{DFA}$, rate of dietary fatty acid oxidation; $Ox_{NEFA}$, rate of whole-body fat oxidation; Ra, appearance rate; TG, triglycerides; VLDL, very low-density lipoproteins.
Values are means with 95% confidence interval (CI) in parentheses for normally distributed data and median [interquartile range] for nonparametric data ($n = 8$).
*Effect of temperature, $P < 0.05$, **$P < 0.01$, ****$P < 0.0001$. ‡In $N = 7$. #Effect of acclimation, $P < 0.05$ (Two-way ANOVA for repeated measures with Bonferonni *post hoc* test).

**Effect of cold acclimation on BAT oxidative metabolism.** Aorta (*input function*) and supraclavicular BAT [11]C radioactivity over time before and after cold acclimation are shown in Fig. 2a,b, respectively. The monoexponential decay slope from tissue peak [11]C activity[11,12], a surrogate of tissue oxidative metabolism, increased 2.6-fold following a 4-week cold acclimation, increasing from 0.015 s$^{-1}$ (95% CI: 0.009–0.020) pre-acclimation to 0.038 s$^{-1}$ (95% CI: 0.023–0.053) post-acclimation (two-tailed paired-sample $t$-test, $P = 0.009$) (Fig. 2c). This represents a 2-fold greater acclimation-induced increase in BAT oxidative metabolism compared with our previous acclimation protocol[10]. Peak BAT [11]C radioactivity, an index of tissue perfusion[13], also increased significantly following 4-week cold acclimation (two-tailed paired-sample $t$-test, $P = 0.05$) (Fig. 2b,d).

**Significant cold-induced fractional and net DFA uptake by BAT.** The relative bio-distribution of orally administered [18]FTHA, 200 min after meal ingestion, before and after a 4-week cold acclimation is shown in Fig. 3f. Of the tissues found in the cervicothoracic region, fractional and net DFA uptake was significantly higher in BAT compared with cervical subcutaneous WAT (scWAT) and skeletal muscles (Fig. 3a,b) (two-way ANOVA for repeated measures with Bonferonni *post hoc* test, $P < 0.004$ versus other tissues). The increase in BAT CT radio-density (in Hounsfield units) throughout the cold exposure (Fig. 3c) suggests that BAT intracellular TGs are the most important source of fuel for thermogenesis in

the postprandial state. The relative DFA uptake by BAT was significantly greater than abdominal scWAT but significantly lower than the myocardium and liver (Fig. 3d). There was no effect of acclimation on relative BAT DFA uptake. The only acclimation-associated difference in relative uptake of DFA was lower post-acclimation DFA uptake in abdominal scWAT (two-tailed paired-sample $t$-test, $P = 0.057$) and visceral WAT (two-tailed paired-sample $t$-test, $P = 0.02$). Organ-specific DFA partitioning as a percentage of DFA partitioning is presented in Fig. 3e. The greatest contributors to DFA clearance were the skeletal muscles and liver, representing 50% (95% CI: 44–57%) and 24% (95% CI: 20–28%). The heart, intra-abdominal and subcutaneous adipose tissues accounted for 2% (95% CI: 2–3%), 15% (95% CI: 12–18%) and 8% (95% CI: 5–12%), respectively, whereas BAT accounted for only 0.3% (95% CI: 0.2–0.5%; two-way ANOVA for repeated measures with Bonferonni *post hoc* test, $P < 0.05$ versus all other organs). There was no effect of acclimation on the organ-specific partitioning of DFA. The relative DFA uptake in BAT was not associated with BAT oxidative index ($\rho = -0.20$, $P = 0.46$) whereas it was associated with BAT blood flow index in the pre-acclimation condition ($\rho = 0.71$, $P = 0.06$), but not post-acclimation ($\rho = -0.18$, $P = 0.71$) (Supplementary Fig. 2).

The relative bio-distribution of [18]FDG after intravenous injection, given immediately after the end of cold exposure, before and after a 4-week cold acclimation of a representative participant is shown in Fig. 4a. Fractional glucose uptake (Fig. 4b)

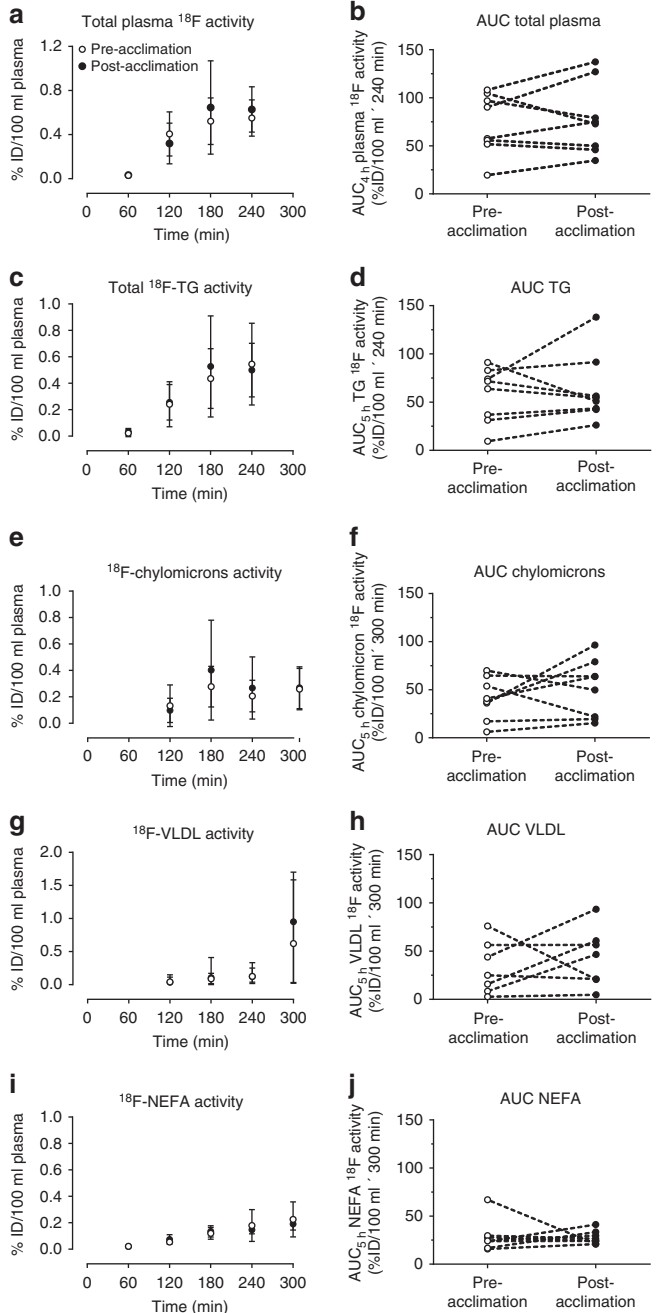

**Figure 1 | Postprandial $^{18}$F activity in circulating lipids.** Total plasma (**a,b**), total TGs (**c,d**), chylomicron (**e,f**), VLDL (**g,h**) ($n = 7$) and NEFA (**i,j**) [$^{18}$F] activity pre- (open circles) and post-cold acclimation (closed circles). %ID, per cent ingested dose. Data are reported as the mean with 95% CI, $n = 8$ (except where indicated).

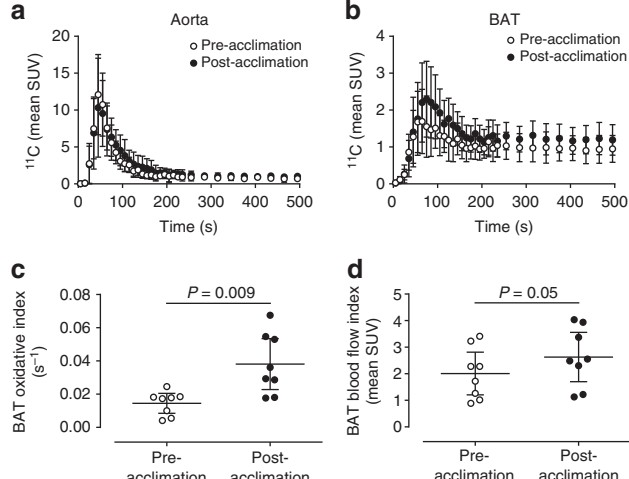

**Figure 2 | [$^{11}$C]-acetate kinetics.** $^{11}$C time-radioactivity curves over the first 500 s of acquisition after [$^{11}$C]-acetate injection before and after cold acclimation in aorta (**a**) and supraclavicular BAT (**b**). Monoexponential decay slope from peak tissue $^{11}$C activity (BAT oxidative index) in supraclavicular BAT (**c**). Peak BAT $^{11}$C activity in supraclavicular BAT (**d**). Data are reported as the mean with 95% CI, $n = 8$. $P$ values are from two-tailed paired-sample $t$-test.

in BAT compared with abdominal scWAT (two-way ANOVA for repeated measures with Bonferonni *post hoc* test, $P < 0.001$) and skeletal muscles (two-way ANOVA for repeated measures with Bonferonni *post hoc* test, $P = 0.004$) but not the myocardium or liver.

## Discussion

Studies performed in mice exposed to the cold or treated with a $\beta_3$-adrenergic receptor agonist have unequivocally demonstrated that the liver and BAT are both equally significant organs involved in metabolizing TG-rich lipoproteins, efficiently clearing TGs from the circulation[4–6]. The present study demonstrates for the first time in humans that BAT can take up DFA during acute mild-cold exposure. However, increasing BAT oxidative capacity 2.6-fold through chronic cold acclimation did not increase DFA uptake any further. In addition, contrary to what was observed in rodent models, BAT DFA partitioning per volume of tissue was ~83% lower than in the liver and 55% lower than in the heart, but was comparable to levels seen in abdominal scWAT and skeletal muscles. Furthermore, BAT only accounted for ~0.3% of total DFA clearance, significantly lower than the contribution of the liver, skeletal muscles, heart and adipose tissues. That DFA partitioning towards BAT did not increase any further following cold acclimation despite a significant increase in BAT oxidative metabolism suggests a limited role of BAT for the clearance of TRL during acute cold exposure.

In the fasted state, BAT thermogenesis is supported in small part by circulating glucose and fatty acids, while fatty acids released from hydrolysis of intracellular TGs appear to be the predominant energy substrate[10,14–16]. The uptake of circulating glucose is further increased following cold acclimation in lean, obese or diabetic individuals[10,17–21]. Here we show for the first time that in the postprandial state, BAT in humans also takes up DFA, while the greater oxidative metabolism following acclimation did not increase DFA BAT uptake any further. Whether DFA were taken up predominantly as a whole TRL particle and/or as fatty acids liberated following

was greater in BAT compared with cervical scWAT (two-way ANOVA for repeated measures with Bonferonni *post hoc* test, $P = 0.04$) and skeletal muscle (two-way ANOVA for repeated measures with Bonferonni *post hoc* test, $P = 0.06$), but there was no difference between pre- and post-acclimation BAT fractional glucose uptake. Net tissue glucose uptake (Fig. 4c) was higher in BAT compared with cervical scWAT (two-way ANOVA for repeated measures with Bonferonni *post hoc* test, $P = 0.02$) and skeletal muscle (two-way ANOVA for repeated measures with Bonferonni *post hoc* test, $P = 0.05$). The relative glucose uptake (Fig. 4d) was significantly greater

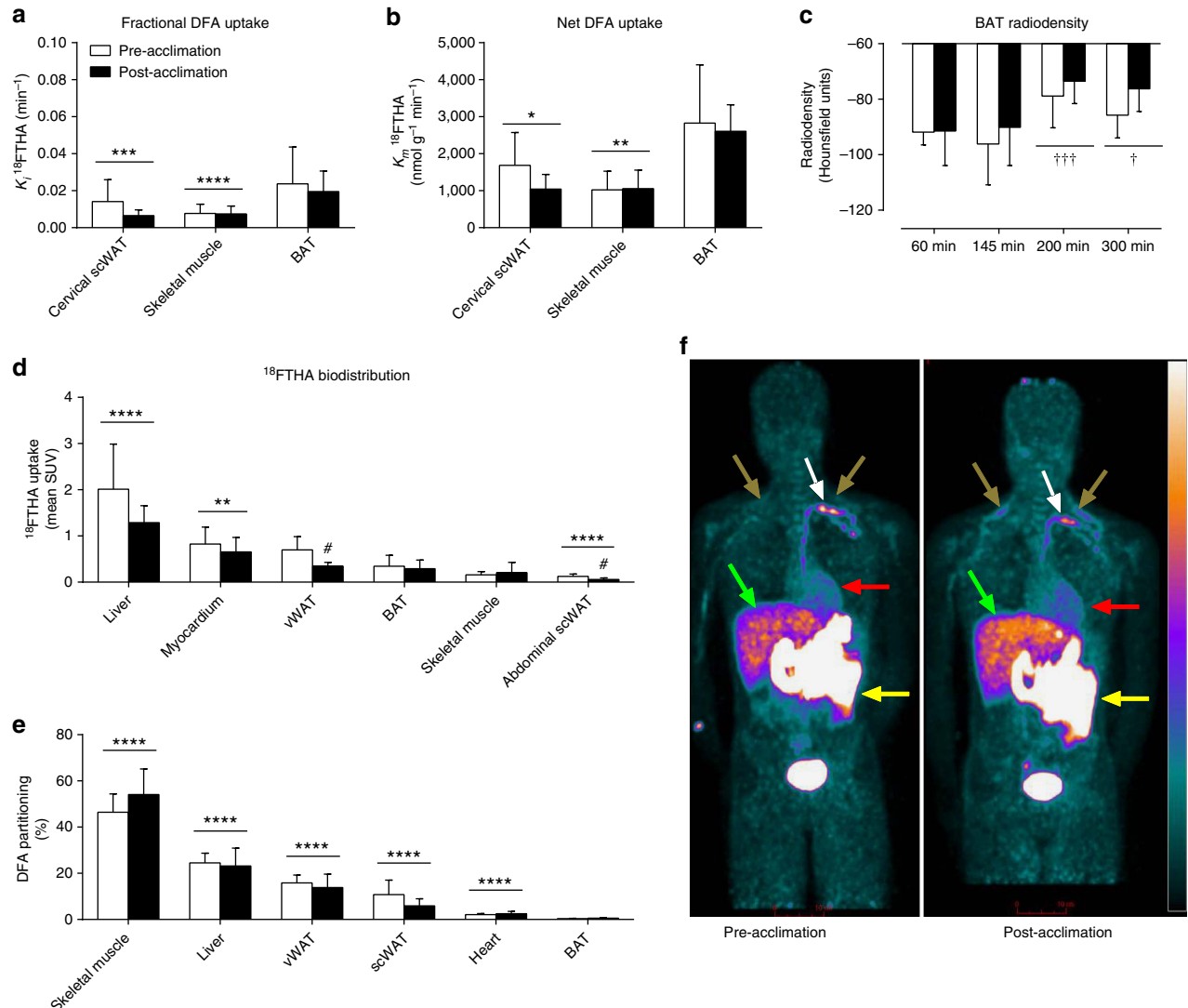

**Figure 3 | Organ-specific partitioning of dietary fatty acids.** Fractional (**a**) and net (**b**) DFA uptake rate in cervical subcutaneous white adipose tissue, skeletal muscles and BAT between 140 and 170 min after oral ingestion of $^{18}$FTHA ($n = 7$). BAT radio-density by CT during cold exposure (**c**). Biodistribution of $^{18}$FTHA following whole-body PET scans per volume of tissue (**d**) and organ-specific contribution to whole-body DFA clearance (**e**). Anterioposterior whole-body PET acquisition performed 200 min after oral ingestion of $^{18}$FTHA pre- and post-acclimation (**f**). Data are reported as the mean with 95% CI, $n = 8$ (except where indicated). ****$P < 0.0001$, ***$P < 0.001$ versus BAT, †††$P < 0.001$, †$P < 0.05$ versus 60 min, #$P < 0.05$ versus pre-acclimation, repeated measures two-way ANOVA with Bonferonni *post hoc* test. White arrows show thoracic duct, brown arrows show supraclavicular BAT, red arrows show myocardium, green arrows show liver and yellow arrows show intestines. Scale bar, 10 cm.

lipoprotein lipase-mediated lipolysis of chylomicrons remains to be determined. Rodent models suggest the latter, as cold-induced increases in gene expression of lipoprotein lipase (*Lpl*) and the long-chain fatty acid transporter cluster of differentiation 36 (*Cd36*) are accompanied by an increase in intravascular lipolysis of circulating TRL, followed by subsequent uptake of liberated fatty acids by BAT[4,5,22]. The significant increase in BAT radiodensity, an indirect marker of decrease in intracellular TG content, combined with the lack of association between BAT oxidative metabolism and uptake of DFA, suggest that postprandial BAT thermogenesis is largely supported by fatty acids hydrolysed from its intracellular lipid pool. DFA may thus likely be used for replenishing this depleting intracellular TG pool. Whether BAT DFA uptake continues to increase following the end of cold exposure is unclear. However, the similar radiodensity during the first hour of acute cold exposure pre- and post-acclimation suggests that the TG pool in

BAT is continually replenished following cold exposure. The time-course and substrates contributing to that replenishment are unknown, but animal models suggest that TRL-derived fatty acids may be the most important source and that BAT activation may not be necessary for this process[7].

The interest in BAT for the treatment of obesity and its associated metabolic complications stems in part from evidence that some diets may elicit a sympathetic response associated with BAT recruitment and thermogenesis to counteract weight gain, a process termed diet-induced thermogenesis[23,24]. Direct evidence supporting a role for BAT in diet-induced thermogenesis in humans has remained scarce and controversial. One recent study demonstrated an increase in BAT glucose uptake following a high calorie meal[25]. However, BAT glucose uptake may be increased under hyperinsulinemic conditions[26], such as after a meal, without increasing BAT thermogenesis. Although the present study cannot discriminate between the sympathetic responses

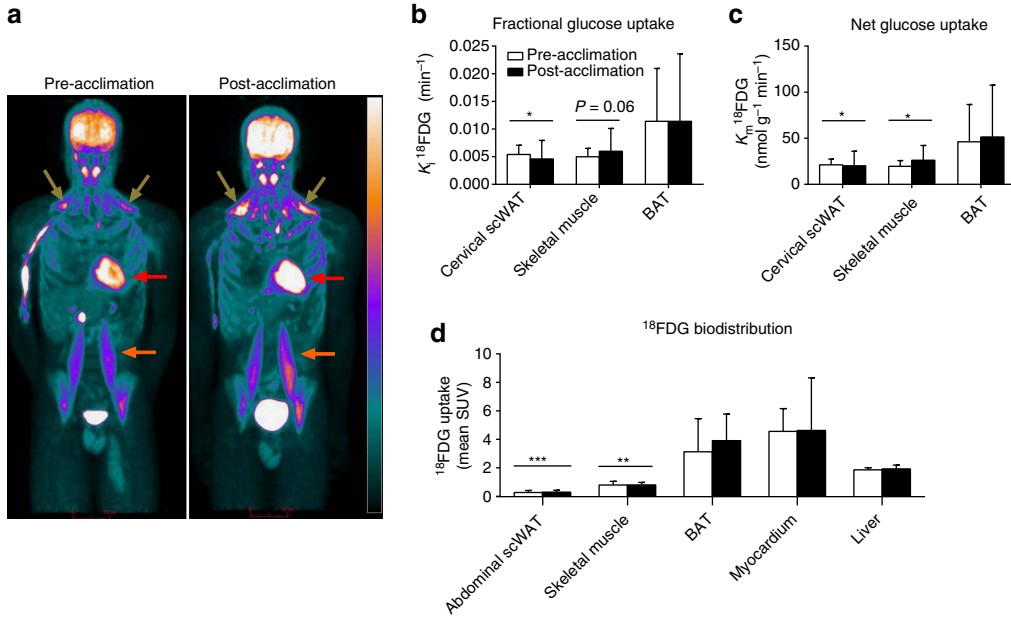

**Figure 4 | Organ-specific partitioning of glucose.** Coronal view (anterior–posterior projection) of whole-body [18]FDG uptake after cold exposure before and after acclimation (**a**). Fractional ($K_i$) (**b**) and net ($K_m$) (**c**) glucose uptake in cervicothoracic tissues ($n = 5$). Biodistribution of [18]FDG following whole-body PET scans (**d**) ($n = 7$). Data are reported as the mean with 95% CI. ***$P < 0.001$, **$P < 0.01$, *$P < 0.05$ versus BAT, Repeated measures two-way ANOVA with Bonferonni *post hoc* test. Brown arrows show supraclavicular BAT, red arrows show myocardium, orange arrows show *m. psoas major*. Scale bar, 10 cm.

attributed to the meal versus cold exposure, the lack of change in BAT DFA uptake in the face of a 2.6-fold increase in BAT oxidative metabolism between pre- versus post-cold acclimation demonstrates that BAT activation has a limited role in dissipating energy directly from DFA in humans.

The present results support the notion that cold exposure stimulates DFA uptake by BAT. At room temperature, we have never previously seen significantly higher uptake of DFA in supraclavicular fat areas compared with neck muscles and other adipose tissue depots (unpublished observations from refs 27–31). When compared with lean healthy men ingesting the same liquid meal at room temperature that we previously reported[27], uptake of DFA by the myocardium, liver and scWAT appears to be lower during cold exposure. Further, the slightly lower [18]F signal in the liver 4 h postprandially combined with the slightly higher [18]F-labelled VLDL TG levels following the 4-week cold acclimation suggests a possible increase in hepatic DFA turn-over after cold acclimation. The accelerated clearance of TRL remnants by the liver following cold acclimation may not be due to BAT clearance of hydrolysed DFA as is the case in rodents. Uptake of DFA by the skeletal muscles was 29% higher following cold acclimation. While not statistically significant when averaging 18 different muscles of varying uptake rates, this difference may have important physiological implications in lean healthy men with a muscle mass representing ∼40% of total body weight and where shivering skeletal muscles may be the primary source of cold-induced heat production.

There are limitations of our study that need discussion. First, previous studies examining the TRL-clearing potential of BAT in rodents were done by injecting TRL-mimicking particles, ensuring 100% bioavailability of TRL in circulation. In contrast, oral administration of our DFA tracer made it only partially available in circulation. Second, BAT metabolism was measured during a relatively short acute cold exposure period in the present study. Perhaps the potential for BAT to clear circulating substrates would be at its highest following

more prolonged cold exposures, when the demand to replenish BAT intracellular lipid pool is expected to be at its highest. The latter would however not explain why BAT DFA uptake was not increased despite significant increase in BAT oxidative metabolism after cold acclimation. It is possible that recruitment of other thermogenic tissues (for example, shivering muscles) may have masked a more important contribution of BAT to DFA clearance. As mentioned above, in the absence of cold-induced BAT metabolic activation, however, we have never seen significant DFA uptake in BAT in our previous studies using the present methods. Future studies using alternative ways than cold exposure to activate BAT metabolism during the postprandial state may help clarify this issue. Finally, 'BAT volume' was not measured in the present study on the basis that there was competition between the uptake of DFA and glucose by the cold-stimulated BAT. The net uptake of glucose in BAT in the present study is a third of what we have previously reported using the same cold stimulus ($51 \, \text{nmol g}^{-1} \text{min}^{-1}$ post-acclimation here versus $215 \, \text{nmol g}^{-1} \text{min}^{-1}$ post-acclimation in ref. 10). Consequently, using typical BAT thresholds (that is, CT radiodensity between $-30$ and $-150$ and mean standard uptake value (SUV) $> 1.5$) would have resulted in a significant underestimation in 'BAT volume'. It is not possible to measure 'BAT volume' using [11]C-acetate due to its very high tissue metabolism or using [18]FTHA due to the poor radioactivity contrast between BAT and WAT.

In conclusion, acute cold exposure leads to BAT DFA uptake in humans. In contrast to what was observed in rodents, BAT DFA uptake is significantly lower than in the liver and myocardium per volume of tissue and contributes only to ∼0.3% of total body DFA clearance. Although cold exposure appears to improve somewhat the systemic clearance of DFA, shivering skeletal muscles display a greater potential than BAT for DFA uptake and metabolism. Despite robust activation of BAT oxidative metabolism after cold acclimation, BAT DFA uptake is not further increased.

## Methods

**Ethical approval.** Informed written consent was obtained from all participants in accordance with the Declaration of Helsinki and the protocol received approval from the Office of Research Ethics and Integrity at the University of Ottawa and the Human Ethics Committee of the Centre de Recherche du Centre Hospitalier Universitaire de Sherbrooke.

**Experimental protocols.** All subjects participated in a 5-h postprandial experimental session before and after a 4-week cold acclimation (Supplementary Fig. 1). The cold acclimation consisted of daily 2-hours cold exposures repeated five consecutive days per week for four consecutive weeks for a total of 18 acclimation sessions. The temperature of the water circulating through the suit during the acclimation sessions was adjusted using a custom-designed closed loop control system to elicit a quick and sustained mean skin temperature of 28 °C (see Supplementary Fig. 1 for example). Such a modification from our previous acclimation protocol[10] was performed to account for the rapid cold habituation that occurred when using a fixed temperature (10 °C) throughout the 4-week acclimation in our previous approach.

Each postprandial acute cold exposure experimental session consisted of a 150 min baseline period at ambient temperature ($\sim$21 °C) followed by 240 min of mild cold exposure, elicited using the liquid-conditioned suit. During cold exposure, 18.0 °C water was circulated through the suit during the pre-acclimation visit and water at a temperature eliciting the same individualized inlet-outlet temperature difference as the pre-acclimation protocol (16.4 °C, 95% CI: 15.6–17.0 °C) was circulated during the post-acclimation visit using a temperature-controlled circulation bath (Isotemp 6200R28, Fisher scientific, USA). The post-acclimation temperature was adjusted to induce a similar thermogenic rate between the two experimental sessions. The same suit was used for all subjects to maintain consistent tubing density and water flow. Experiments were conducted between 07:30 and 17:00 hours, following a 12 h fast and 48 h without strenuous physical activity. Subjects were asked to follow a two-day standard isocaloric diet based upon a 3-day food record, filled a validated questionnaire for physical activity[32] and underwent portable arm band accelerometry for 7 days[33]. Upon their arrival in the laboratory, subjects wearing only shorts were weighed and instrumented with autonomous wireless temperature sensors (Thermochron iButton model DS1922H, Maxim) placed on the forehead, upper back, lower back, abdominals, forearm, quadricep, hamstring, back calf, anterior tibialis, chest, bicep and hand to measure mean skin temperature[34]. Surface electromyography (EMG) electrodes (Delsys, EMG System, USA) were placed on the belly of eight muscles: *m. pectoralis major*, *m. deltoideus*, *m. trapezius*, *m. sternocleidomastoid*, *m. bicep brachii*, *m. rectus femoris*, *m. vastus medialis* and *m. vastus lateralis*. Participants were then fitted with the liquid-conditioned suit, swallowed a telemetric thermometry capsule to measure core temperature (Vital Sense monitor and Jonah temperature capsule, Mini Mitter, Bend, OR, USA) and performed a series of maximal voluntary contractions (MVC) of each of the 8 muscles being recorded by EMG for normalization of the shivering measures. Indwelling catheters were then placed in an antecubital vein in both arms for blood sampling and tracer infusions. Participants were asked to empty their bladder and a primed ($3.3 \times 10^6$ dpm min$^{-1}$) continuous infusion ($0.33 \times 10^6$ dpm min$^{-1}$) of [3-$^3$H]-glucose was started at $-150$ min to determine the plasma glucose appearance rate (Ra$_{\text{glucose}}$)[35]. A continuous infusion of [7,7,8,8-$^2$H$_4$]-potassium palmitate (0.01 μmol kg$^{-1}$ min$^{-1}$ in 100 ml 25% human serum albumin) and a primed (1.6 μmol kg$^{-1}$) continuous infusion (0.08 μmol kg$^{-1}$ min$^{-1}$) of [1,1,2,3,3-$^2$H]-glycerol were started at $-120$ min to determine plasma NEFA appearance rate (Ra$_{\text{NEFA}}$) and plasma glycerol appearance rate (Ra$_{\text{glycerol}}$)[36].

Upon the start of cold exposure ($t = 0$ min), participants ingested a standard liquid meal[37], mixed with 3.6 μmol kg$^{-1}$ of [U-$^{13}$C]-palmitate (preceded by a 1.2 μmol kg$^{-1}$ intravenous bolus of [1-$^{13}$C]-NaHCO$_3$ given at $-120$ min) and $\sim$70 MBq of $^{18}$FTHA mixed into Intralipid 20% (Baxter, Mississauga, Ontario, Canada) and inserted into gel capsules (T U B Enterprises) to determine the oxidation rate of ingested dietary fat (Ox$_{\text{DFA}}$)[38] and whole-body DFA partitioning[3,39], respectively. The standard liquid meal was prepared by sonication of soybean oil (42 g l$^{-1}$), canola oil (35 g l$^{-1}$), dried non-fat milk (263 g l$^{-1}$), egg phospholipids (0.18 g l$^{-1}$) and water with the addition of chocolate syrup (150 ml l$^{-1}$) to provide 2133 kcal l$^{-1}$. Participants ingested 400 ml of this liquid meal over 20 min providing a total of 853 kcal, consisting of 31 g or 33% as fat, 42 g or 18% as proteins and 101 g or 49% as carbohydrates. Shivering EMG signals were continuously recorded during cold exposure and corrected for baseline muscle EMG activity and normalized for a MVC (%)[40]. In brief, raw EMG signals were collected at 1,000 Hz, filtered to remove spectral components below 20 Hz and above 500 Hz as well as 60- Hz contamination and related harmonics, and analysed using custom-designed MATLAB algorithms (Mathworks, Natick, MA). Shivering intensity of individual muscles was determined from root mean square (r.m.s.) values calculated from raw EMG. In short, baseline RMS values (RMS$_{\text{baseline}}$: 5 min RMS average measured before cold exposure) were subtracted from shivering RMS (RMS$_{\text{shiv}}$) values and RMS values obtained from the MVCs of individual muscles (RMS$_{\text{mvc}}$). Shivering intensity was determined by using a weighted mean of the shivering intensity of all 8 muscles. A shivering burst was defined as an EMG interval with a duration $> 0.2$ s, an

inter-burst interval $> 0.75$ s and an amplitude higher than the intensity threshold at each recording period. Intensity threshold was determined by: (i) averaging shivering intensity ($A_{\text{EMG}}$) over the entire recording period, (ii) averaging the remaining values above $A_{\text{EMG}}$ ($B_{\text{EMG}}$) and (iii) setting the intensity threshold at $B_{\text{EMG}}$. Whole-body metabolic heat production was determined by indirect respiratory calorimetry, corrected for protein oxidation[41].

Tissue oxidative metabolism was determined 60 min into cold exposure by first performing a segmental CT scan (40 mAs) centred at the cervicothoracic junction to correct for attenuation, measure BAT radiodensity and define PET regions of interest. A $\sim$185 MBq bolus of $^{11}$C-acetate was then injected intravenously and followed by a 30- min list-mode dynamic PET acquisition ($24 \times 10$ s, $12 \times 30$ s and $4 \times 300$ s)[14]. Tissue oxidative metabolism index (the rapid fractional tissue clearance of $^{11}$C-acetate, $k$, in s$^{-1}$) was estimated from tissue $^{11}$C activity over time using monoexponential fit from the time of peak tissue activity[42]. Determination of tissue oxidative metabolism using the $^{11}$C-acetate method is based on the following assumptions[43]: (a) acetate enters the Krebs cycle freely after rapid conversion into acetyl-CoA; (b) other acetate metabolic fates (for example, *de novo* lipogenesis) are relatively slow compared with the Krebs cycle carbon fluxes; (c) carbon fluxes into the Krebs cycle through acetyl-CoA are directly coupled to the production of reducing equivalents; (d) the Krebs cycle contribution to the production of reducing equivalents is stable and accounts for approximately two thirds of total production and (e) the production of reducing equivalents is tightly coupled to oxygen consumption. The time-activity curve from the cervical scWAT (Supplementary Fig. 4) illustrates a tissue taking up $^{11}$C-acetate but without a rapid clearance (indicating non-oxidative disposal), in contrast to Fig. 2b demonstrating a rapid tissue clearance of $^{11}$C-acetate following its extraction from circulation (indicating oxidation). After a segmental CT (40 mAs), to measure BAT radiodensity and for definition of PET regions of interest, dynamic list-mode PET acquisition centred at the cervicothoracic junction was performed between time 140 and 170 min into cold exposure ($15 \times 120$ s) to determine oral $^{18}$FTHA uptake rate in BAT, cervical scWAT and three skeletal muscles (*m. deltoideus*, *m. trapezius*, *m. pectoralis major*) using a Patlak linearization method[44], with the image-derived arterial input function taken from the aortic arch[45]. At time 200 min, a whole-body PET/CT acquisition was performed to examine the biodistribution of $^{18}$FTHA. The $^{18}$FTHA biodistribution data, expressed as SUV, is a measure of the relative DFA uptake and retention by a tissue compared with the rest of the body. DFA partitioning was presented both per volume of tissue (Fig. 3d) and its relative contribution to whole-body clearance according to the volume of distribution of the absorbed DFA (Fig. 3e). The latter assumes that: (1) total volume of distribution of absorbed DFA (that is, after DFA are packaged into chylomicron-TG and secreted in the systemic circulation) can be approximated by the sum of the volumes of the liver, skeletal muscles, intra-abdominal and subcutaneous adipose tissues, myocardium and BAT[27]; (2) skeletal muscle mass accounts for 52% of lean mass in men[46] and a tissue density of 1.112 g ml$^{-1}$ (ref. 47); (3) intra-abdominal adipose tissue accounts for 25% of total body adipose tissue in men and a tissue density of 0.9196 g ml$^{-1}$ (ref. 48); (4) subcutaneous adipose tissue accounts for 75% of total body adipose tissue in men and a tissue density of 0.9196 g ml$^{-1}$ (ref. 48); (5) myocardial mass accounts for 0.4% of body weight[49] and a tissue density of 1.053 g ml$^{-1}$ (ref. 50); and (6) an average BAT mass of 100 g in lean healthy men[51], which increases by 45% following cold acclimation[10,19,20], and a tissue density of 0.9196 g ml$^{-1}$. Liver volume was determined by drawing the liver surface on each 5 mm slice of the whole-body CT scan (16 mAs) at time 300 min and reconstructed in 3D volume using OsiriX Lite 8.0.1 software (Pixmeo SARL, Bernex, Switzerland)[31]. After a segmental CT (40 mAs), an intravenous bolus of [$^{18}$F]-fluorodeoxyglucose ($^{18}$FDG) ($\sim$185 MBq) was given at time 240 min followed by a 30- min list-mode dynamic PET acquisition ($12 \times 10$ s, $8 \times 30$ s, $6 \times 90$ s and $4 \times 300$ s) centred at the cervicothoracic junction to determine tissue-specific glucose uptake using Patlak linearization method[44], with the image-derived arterial input function taken from the aortic arch[45]. Tissue and blood radioactivity from oral $^{18}$FTHA was subtracted for these analyses, as we previously showed $^{18}$F tissue levels reaching a plateau in all organs after 180 to 240 min[27]. Tissue net fatty acid and glucose uptake ($K_m$) were then calculated by multiplying $K_i$ by plasma chylomicron and glucose concentrations, respectively, measured during the PET imaging protocol. Following cold exposure, a whole body PET acquisition was performed to determine $^{18}$FDG organ distribution and tissue SUV. One participant could not receive the $^{18}$FDG injection during one of the studies. Therefore $N = 7$ participants are reported for outcomes based on $^{18}$FDG biodistribution. Three participants were not able to receive an $^{18}$FDG PET dynamic acquisition, due to participants moving following static $^{18}$FTHA PET scan.

$^{18}$FTHA metabolites were determined using a solid-phase extraction method[27]. Briefly, Oasis MAX 6-ml (500 mg) LP extraction cartridges (Waters, Mississauga, ON, Canada) were activated with 6 ml of acetonitrile (ACN), 6 ml of methanol (MeOH) and 6 ml of 2% ammonium hydroxide (NH$_4$OH–H$_2$O). This solution was applied to the cartridge and the liquid phase that contains TG was pulled through and collected. The column was then washed with 10 ml of 2% NH$_4$OH–H$_2$O–70% ACN–MeOH to remove the remaining TGs, which were collected in the same tube. The column was then eluted with 10 ml of 5% formic acid–70% ACN–MeOH to collect NEFA into another tube. TG and NEFA samples were transferred into a counting γ-tube to measure $^{18}$F activity. Chylomicrons were separated in Quick-Seal centrifuge tubes from 2 ml of heparinized plasma gently overlaid with 3.5 ml of

distilled water centrifuged for 20 min at 33,000 r.p.m. at 20 °C (Optima L-100XP ultracentrifuge with 100Ti Beckman rotor)[52]. After ultacentrifugation, the chylomicron fraction floating at the top of the tube was removed ($\sim$1.5 ml) and 700 µl was transferred into a counting γ-tube to measure $^{18}$F activity. The remaining liquid ($\sim$4.0 ml) was transferred to a separate Quick-Seal centrifuge tube gently overlaid with 1.5 ml of distilled water and centrifuged for 90 min at 82,400 r.p.m. at 20 °C (Optima L-100XP ultracentrifuge with 100Ti Beckman rotor)[53,54]. After ultacentrifugation, the VLDL fraction floating at the top of the tube was removed ($\sim$1.5 ml) and 700 µl was transferred into a counting γ-tube to measure $^{18}$F activity.

**Biological assays.** Glucose, insulin, total NEFA TG, plasma cortisol, TSH, free T3 and free T4 were measured using specific radioimmunoassays and colorimetric assays[14,36]. Plasma C-peptide, GIP, total GLP-1, glucagon, insulin and leptin and PYY were measured using Luminex xMAP-based immunoassays (Millipore, Etobicoke, ON, Canada). ACTH, GH, adiponectin and acylated ghrelin were measured by ELISA (Alpco, Salem, NH, USA). Individual plasma NEFA (palmitate, linoleate, oleate), [U-$^{13}$C]palmitate enrichment, [7,7,8,8-$^2$H$_4$]palmitate enrichment and [1,1,2,3,3-$^2$H]glycerol enrichment were measured by gas chromatography–mass spectrometry[36]. [3-$^3$H]-glucose specific activity was determined by liquid scintillation spectrometry[35].

**Statistical analysis.** The sample size was based on the number of subjects necessary to determine 4-week cold acclimation-stimulated BAT oxidative metabolism. Our previous study showed that $N = 6$ subjects confers a power of >80% to detect significant increases in BAT oxidative metabolism using the $^{11}$C-acetate PET method at a two-sided alpha level of 0.05 (ref. 10). Data are expressed as mean with 95% CI. Paired Student's $t$-test was used to compare between acute cold exposure experimental sessions. Two-way ANOVA for repeated measures with acclimation status, acute cold exposure/postprandial state and their interaction as the independent variables was used to analyse acclimation- and temperature-dependent differences. Bonferonni's multiple comparisons post hoc test was used, where applicable. Appropriate transformations of variables were performed when normal distribution was not observed for parametric statistical testing. A two-tailed $P$ value of <0.05 was considered significant. All analyses were performed using SPSS for Windows (version 21.0; SPSS, Chicago, IL) or GraphPad Prism version 6.0 for Windows (GraphPad, San Diego, CA).

**Data availability.** The data that support the findings of this study are available from the corresponding authors (A.C.C. and F.H.) upon reasonable request.

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

## Acknowledgements

The authors would like to acknowledge the excellent technical assistance provided by Diane Lessard, Caroll-Lynn Thibodeau, Maude Gérard, Éric Lavallée, Lucie Bouffard, Mélanie Fortin, Olivier Mantha-Landry, Sophia Raytchev and Frédéric Tassé. This work was supported by an operating grant from the Canadian Institutes of Health Research (CIHR) to A.C.C. and the Natural Sciences and Engineering Research Council of Canada (NSERC) to F.H. D.P.B. and C.N. are the recipients of a CIHR Postdoctoral fellowship. A.C.C. is the recipient of the CIHR-GlaxoSmithKline Chair in Diabetes. The Centre de Recherche du Centre hospitalier universitaire de Sherbrooke is funded by the Fonds de Recherche du Québec – Santé. We also thank the participants of this study for their collaboration and Allen-Vanguard (Kevin Semeniuk) for providing the liquid-conditioned suits.

## Author contributions

Conception and design of the experiments: A.C.C., D.P.B., F.H., D.R., E.E.T. and B.G. Collection, analysis and interpretation of data: D.P.B., H.C.T., C.N., F.F., S.P., F.H., E.E.T., D.R. and A.C.C. Drafting the article or revising it critically for important intellectual content: D.P.B., A.C.C., F.H., D.R., E.E.T. and B.G.

## Additional information

**Competing financial interests:** The authors declare no competing financial interests.

