## [Peer Review File · Nature Communications]

Reviewers' comments:

Reviewer #1 (Remarks to the Author):

The authors have made extensive and appreciated efforts to respond to the original set of comments. All are satisfactory with one exception. Comment "B" asked the authors to calculate the BAT volume and then report the DFA uptake per mL of BAT, which would give a specific estimate of how much BAT mass or relative activity would be required to treat dyslipidemia. Their response, which was added to the end of the limitations section, substantially improves the discussion of the matter.

However, it would be helpful if the authors could provide the rates of DFA uptake normalized to some feature of the BAT that can also be applied to the WAT and skeletal muscle. This would improve the current study and also provide much-needed data for the field since this information would be derived from human studies rather than in vitro or rodent models whose translatability is unknown. It is agreed that BAT volume will not be as useful in the current study since it will be underestimated. The authors should provide, if they have it, graphs for WAT and skeletal muscle similar to the ones for BAT shown in Supplemental Figure 2.

It is good to see that the authors they removed the final sentences of the Abstract and Discussion.

p. 5, line 21 has a dangling phrase "no effect...uptake" that needs to be addressed.

We provide below a point-by-point response to each comment and a modified version of our manuscript.

Reviewers' Comments:

The authors have made extensive and appreciated efforts to respond to the original set of comments. All are satisfactory with one exception. Comment “B” asked the authors to calculate the BAT volume and then report the DFA uptake per mL of BAT, which would give a specific estimate of how much BAT mass or relative activity would be required to treat dyslipidemia. Their response, which was added to the end of the limitations section, substantially improves the discussion of the matter.

However, it would be helpful if the authors could provide the rates of DFA uptake normalized to some feature of the BAT that can also be applied to the WAT and skeletal muscle. This would improve the current study and also provide much-needed data for the field since this information would be derived from human studies rather than in vitro or rodent models whose translatability is unknown. It is agreed that BAT volume will not be as useful in the current study since it will be underestimated.

Thank you for this valuable feedback. In the first round of revisions, we added a panel to Figure 3 (Figure 3 B) which reports net DFA uptake relative to tissue mass (per gram of tissue) for skeletal muscle, subcutaneous WAT and BAT in the cervicothoracic region. Upon further reflection and thanks to this reviewer's comments, we decided to review our whole-body DFA uptake data and to calculate the organ-specific DFA uptake relative to whole-body clearance according to the volume of distribution of the absorbed DFA (Figure 3E). This approach makes the following assumptions:

- (1) Total volume of distribution of absorbed DFA (i.e. after DFA are packaged into chylomicron-TG and secreted in the systemic circulation) can be approximated by the sum of the volumes of the liver, skeletal muscles, intra-abdominal and subcutaneous adipose tissues, myocardium and BAT (PMID: 21098737).
- (2) Skeletal muscle mass accounts for 52% of lean mass in men (PMID: 12145010) and a tissue density of 1.112 g/mL (PMID: 16154420);
- (3) Intra-abdominal adipose tissue (vWAT) accounts for 25% of total body adipose tissue in men and a tissue density of 0.9196 g/mL (PMID: 18175737);
- (4) Subcutaneous adipose tissue accounts for 75% of total body adipose tissue in men and a tissue density of 0.9196 g/mL (PMID: 18175737);
- (5) Myocardial mass accounts for 0.4% of body weight (PMID: 9234964) and a tissue density of 1.053 g/mL (PMID: 14693681);

(6) An average BAT mass of 100 g in lean healthy men (DOI: 10.2217/clp.15.14), which increases by 45% following cold acclimation (PMID: 24423363, 23867626, 24954193), and a tissue density of 0.9196 g/mL.

We also directly measured the liver volume of the participants by drawing the liver surface on each 5 mm slice of the whole-body CT scan at time 300 min after meal intake with reconstruction in 3D using OsiriX Lite 8.0.1 software (Pixmeo SARL, Bernex, Switzerland), as previously described (PMID: 26224886).

We now report the contribution of each of these organs to DFA clearance on page 6:

*'Organ-specific DFA partitioning in % of total DFA partitioning is presented in **Figure 3E**. The greatest contributors to DFA clearance were the skeletal muscle and liver, representing 50% (95%CI: 44 to 57%) and 24% (95% CI: 20 to 28%). The heart, intra-abdominal and subcutaneous adipose tissues accounted for 15% (95%CI: 12 to 18%), 8% (95% CI: to %) and 8% (95% CI: 5 to 12%), respectively, whereas BAT accounted for only 0.3% (95% CI: 0.2 to 0.5%, $P < 0.05$ vs. all other organs). There was no effect of acclimation on the organ-specific partitioning of DFA.'*

The authors should provide, if they have it, graphs for WAT and skeletal muscle similar to the ones for BAT shown in Supplemental Figure 2.

We have added these panels to Supplemental Figure 2.

p. 5, line 21 has a dangling phrase “no effect...uptake” that needs to be addressed.

We have corrected this dangling phrase as part of our greater correction.

REVIEWERS' COMMENTS:

Reviewer #1 (Remarks to the Author):

The authors have made a very nice effort to response to the remaining comments with quantitative detail.